# A Qualitative Analysis of Motivators to Participation in Suicide-Focused Research from a Community-Based Australian Sample

**DOI:** 10.3390/ijerph18094705

**Published:** 2021-04-28

**Authors:** Demee Rheinberger, Fiona Shand, Katherine Mok, Lauren McGillivray, Myfanwy Maple, Alexander Burnett, Lisa N. Sharwood, Nicola A. Chen, Michelle Torok

**Affiliations:** 1Black Dog Institute, University of New South Wales, Hospital Road, Randwick, NSW 2031, Australia; fionas@unsw.edu.au (F.S.); katherinem12@gmail.com (K.M.); l.mcgillivray@blackdog.org.au (L.M.); alexander.burnett@blackdog.org.au (A.B.); m.torok@unsw.edu.au (M.T.); 2Faculty of Medicine and Health, University of New England, Armidale, NSW 2351, Australia; mmaple2@une.edu.au; 3Faculty of Medicine and Health, The University of Sydney, Camperdown, NSW 2006, Australia; l.sharwood@unsw.edu.au; 4Faculty of Engineering and Risk, University of Technology Sydney, Ultimo, NSW 2007, Australia; 5Orygen, Parkville, VIC 3052, Australia; nicola.chen@unimelb.edu.au; 6Centre for Youth Mental Health, University of Melbourne, Parkville, VIC 3052, Australia

**Keywords:** suicide, suicide prevention, research participation, lived experience, thematic network analysis, qualitative methods

## Abstract

Suicide prevention strategies internationally appear to be falling short of making a meaningful impact on global suicide deaths. Increasing the rates of general community participation in suicide research may improve knowledge generalisability as it relates to suicidal behaviour and leads to new suicide prevention approaches. This study aims to explore the motivations of a community-based sample to participate in suicide research. A subsample of the Australian general population took part in an online survey which is part of a multilevel suicide prevention trial. The survey concluded with an optional open-text question asking about peoples’ motivations for participating in the study; 532 participants left a response to this question. These responses were qualitatively analysed using Thematic Network Analysis. Motivations to participate in suicide research were represented by four global themes: altruism, solve systemic problems, lived experience, and personal benefit. Of these themes, three were focused on the benefit of others, while only the final theme articulated motivation to participate that was self-focused. The impact of suicide is felt throughout the wider community. This new understanding of the motivations of community-based samples to participate in suicide research should be used to increase participation rates and reach people who would not normally contribute their voice to suicide research.

## 1. Introduction

Suicide will affect most Australians at some point in their lives and has long-lasting impacts. Despite some countries making progress toward suicide prevention, globally, there has been no significant reduction in the number of suicide deaths over the last few decades [1]. The absence of any significant decline in suicide rates suggests that suicide prevention efforts are falling short of achieving meaningful impact. In part, this may be attributable to gaps in our knowledge of suicide behaviours. To address gaps in understanding why current prevention approaches may not be working, and to identify other targets for prevention, it is necessary to engage a broader reach of community populations in research. By increasing community participation in suicide prevention research, study findings will have greater external validity which may enable us to improve and refine our suicide prevention efforts. To access community-based populations, traditional methods involving postal and telephone surveys are being supplanted by online approaches as we see declines in the usage of landline telephone connections [2], increased privacy restrictions limiting access to electoral roll data, and increasing postage costs [3]. While online surveys have been shown to fail to engage subsamples of the general population, such as participants with lower socioeconomic status, due, in part, to computer illiteracy or poor access to technology [4,5], online surveys have been shown to have similar representativeness to traditional research methods in respect to age, gender, education, and cultural diversity and are effective for reaching traditionally hard-to-reach populations [3], and, often at significantly lower financial and time costs for both participants and researchers [3,6,7]. However, general community willingness to participate in surveys broadly has been declining [8]. Decreased participation in surveys has been attributed to an increase in the number of surveys being conducted, a decrease in free time, and a distrust of survey research [9]. Understanding motivations of those who do self-select to participate in online surveys is essential to improving participation rates, by ensuring that the design of suicide prevention research is relevant, inclusive, and acceptable to communities of interest. However, we do not yet fully understand what motivates participation in online suicide research, as few studies have used this methodology in the area of suicide prevention [7,10,11,12].

Motivation research to date has largely focused on exploring motivations to participate in face-to-face suicide and mental health research [13,14,15,16]. Altruism appears to be a key reason that individuals gave for participation in this context, with studies reporting up to 85% of participants indicated a desire to help, or be useful to others, as a factor driving their participation [11,14,16,17]. Curiosity and a desire to learn also appear to play a role, for example, Dyregrov and colleagues [14] interviewed individuals bereaved by suicide and found that 20% reported that their motivation was the opportunity to gain insight into suicide in general, or the suicide of a loved one. Similarly, an opportunity to learn about one’s own mental health [16], or the opportunity to be heard, vent, and share experiences have also been recorded as motivators for participation in mental health research [14]. Participants indicated being able to discuss personal suicide experience with researchers in a non-judgemental and open setting was important for individuals with current suicidal thoughts [11]. 

Previous studies exploring motivations for participation in suicide research have focused on individuals with lived experience as they are the closest “source of truth” on the experience of suicide. However, for every suicide that occurs, it is estimated that an additional 135 people are impacted or exposed [18], showing that suicide can have wide-reaching social impacts that extend into the individual’s larger community [19,20]. Given certain populations, such as men, older adults, and culturally and linguistically diverse minority groups, are less likely to participate in research [21], but have a higher risk of suicide [22,23], it is important to implement strategies which are likely to encourage participation by a generalised sample of the community. Few suicide prevention studies have been conducted on cross-sectional community samples [7,12,20,24,25], and very few have been able to recruit generalisable samples [24]. Digital research helps researchers to connect with individuals that cannot be reached through the usual methods of recruitment, such as convenience sampling of individuals in support groups or those accessing mental health services. Therefore, the community represents an important source of information for researchers who are building new ways of exploring and understanding suicide to guide suicide prevention strategies. 

While there is research on motivations to participate in suicide research for individuals with lived experience, the authors are not aware of any research looking at the motivations for large-scale, community-wide participation in digital suicide research. This study aims to explore the motivations for participation in a community-wide digital suicide prevention survey through the qualitative analysis of open-text participant responses. Understanding participant motivation has important implications for the design of future studies and recruitment strategies to optimise participation and increase the breadth and richness of information collected via online survey methods.

## 2. Materials and Methods

### 2.1. Study Setting

This cross-sectional survey is part of a stepped-wedge multilevel suicide prevention trial conducted by Black Dog Institute [26] across four New South Wales (NSW) trial regions (Newcastle, Central Coast, Illawarra Shoalhaven, and Murrumbidgee) and three control regions (South Western Sydney, Western NSW, and Nepean Blue Mountains). This study was approved by the Hunter New England Human Research Ethics Committee (HREC/16/HNE/399). No identifiable information provided by participants were included in the reporting of the findings. 

### 2.2. Participants

Study participants were members of the general community recruited online via Facebook and Instagram advertisements [7]. Advertisements were both gender-neutral and male-targeted. The gender-neutral adverts contained phrases such as “Lend a Voice to Suicide Prevention” and gender-neutral imagery. Male-targeted adverts displayed male-focused imagery, used male pronouns, and contained phases such as “voice of local blokes to help make it better”. Ads were targeted at individuals who met the inclusion criteria: aged 18 years or over and living in a trial or control region of the NSW LifeSpan suicide prevention trial. More information about the advertisements and the conversion rates of the advertisements can be found in Lee and Torok et al. [7]. Participant information statements were provided electronically. Participants were informed that they consented to participating by beginning the survey. Individuals not eligible to participate were those aged <18 years (self-report), or those residing outside trial regions. Analysis conducted be Lee and Torok et al. [7] found that the sample was not representative of the wider community.

### 2.3. Data Collection

Data were collected between May 2017 and January 2019. To account for sample attrition, a second sample of participants was also recruited at the end of the first year of the trial for NSW trial regions (May 2018 to March 2019). After clicking the online advertisement, individuals were directed to the survey hosted on Qualtrics. The survey was designed to measure change in suicide literacy, stigma, and behaviours in the community over the course of the LifeSpan trial. The survey consisted of 40 questions in total, which included demographic characteristics and a number of scales measuring suicide-related risk factors—including psychological distress, suicide literacy, and stigma of suicide—and a final open-ended question which asked participants “Do you have any comments or feedback to add (e.g., your motivations for participating in this research, your experience participating in this research, your thoughts on the survey)?”. The data used for analysis in this study were the qualitative responses to this final question addressing motivation for participating. 

### 2.4. Procedure

A total of 8533 participants completed the survey (77.8% of all survey attempts); and 2241 (26%) left a comment in the optional open-text field. Initial content analysis was then conducted by the lead author (D.R.) to separate comments addressing motivation from the wider pool of responses. All responses to the open-text question were categorised into two groups—motivation or not motivation—based on the following criteria:Motivation—included responses that explicitly indicated motivation or decision to participate, such as “my motivation was”, “I participated because”, “I chose to do the survey because”.Not Motivation—responses that did not mention motivation explicitly (i.e., referenced other aspects of the prompt), such as suggested changes to survey structure, participants further explaining their responses or stance on suicide, and/or comments of commendation and support to researchers.

To corroborate the content analysis conducted, a randomly generated subsample of 241 responses (11%) was provided to co-author (N.A.C.) for content analysis. There was an 81% agreement between authors, indicating good interrater reliability [27]. For the 19% of responses where agreement was not reached, a third researcher (A.N.) undertook independent coding to resolve discrepancies. A sample of 534 motivation comments were included for analysis, although two comments were recategorized as not motivation during the coding phase (with agreement from authors N.A.C. and L.M.). The final sample included 532 motivation comments (Figure 1).

Prior to data analysis, the authors (D.R., L.M., and N.A.C.) underwent a reflexive exercise guided by Mauthner and Doucet [28], which involved group discussions about biases and lived experience so we were able to mindfully consider this during the data analysis. 

### 2.5. Data Analysis

Demographic and mental health characteristics were analysed using IBM SPSS statistical software version 25.0 (IBM Corp., Armonk, NY, USA) and qualitative data using Nvivo 12 (QSR International, Burlington, MA, USA). Data were treated as missing if the participant did not provide a response. The representativeness of our sample was assessed by comparing characteristics of those who provided a motivational open-text response (*n* = 532) with the total survey cohort (*n* = 8533). Specifically, the characteristics of gender, age, relationship status, employment status, highest level of education, and history of mental health diagnosis were assessed using the Chi-square goodness-of-fit statistic to determine whether those who left a motivational open-text response were consistent with the total survey cohort. The alpha value was set at 0.05 for all quantitative analyses. 

All open-text responses (*n* = 2241) were analysed using content analysis within a grounded theory framework. The final sample (*n* = 532) of motivation responses were analysed using the 6 steps of Thematic Network Analysis (TNA) [29], see Figure 2, using Nvivo 12. Responses were initially coded by the lead author and codes were reviewed and refined by a second author (L.M.) to organise the responses into both organising and global themes. After the responses were organised into themes, narrative results were synthesised and verbatim quotes from participants were included to illustrate the basic themes.

## 3. Results

### 3.1. Participant Demographics and Representativeness 

Table 1 shows the characteristics of participants who left a motivation comment compared with the total sample. Of the total sample of participants (*n* = 8533), 67.2% were female with an average age of 43.8 years (SD = 14.0, range 18–98). Most participants (*n* = 5674, 66.8%) had been diagnosed with a mental health condition at some point in their lives, were educated to an undergraduate level (*n* = 5343, 62.9%), were employed full-time (*n* = 3836, 45.3%), and 57.6% were in a domestic partnership (either married or de facto, *n* = 4871). The sub-sample of participants had an average age of 45.1 years (SD = 14.5, range = 18–86). Majority of participants identified as female (*n* = 344, 65.0 %), had been diagnosed with a mental health condition at some point in their life (*n* = 384, 72.2%), and had some level of undergraduate education (*n* = 327, 61.7%). Most participants were employed full time (*n* = 220, 41.4%), and 60.3% were in a domestic partnership (*n* = 321). 

There were no significant differences between all participants and those in the sub-sample of 532 participants who had left a comment related to motivation, except for mental health diagnosis. A significant association between participants who left a motivation-related comment and previous mental-health-related diagnosis (χ^2^ (1, *n =* 9025) = 6.55, *p* = 0.01), with a higher proportion of mental health diagnosis among participants who left a motivation-related comment (*n* = 384, 72.2%) compared to the total sample (*n* = 5674, 66.8%).

### 3.2. Thematic Network Analysis

In the first instance, the motivation responses were consolidated into 13 organising themes. Two potential global themes were apparent during the initial coding phase: “Altruism” and “Lived Experience”. These global themes were retained and two additional global themes, “Personal Benefit” and “Solve Systemic Problems”, were identified. Global and organising themes were reviewed by L.M., resulting in one organising theme and the two basic themes within being removed and some responses being moved into other categories. Responses were coded multiple times if they discussed more than one of the themes. The final thematic networks comprised of 36 basic themes, 12 organising themes, and four global themes, which are presented in Table 2.

Four global themes were identified, of which three—altruism, solve systemic problems, and lived experience—focused on motivations that serve external factors (e.g., other people or the mental health care system) and one—personal benefit—on individual motivation (e.g., only of benefit to the individual participant). 

#### 3.2.1. Altruism

Altruism, the desire to increase the welfare of another [30], was frequently recorded as a motivator for participation in the LifeSpan survey. Motivation was largely driven by the statement that partaking in suicide research would serve as a catalyst for positive change.

Participants’ responses indicated a desire to help other people was a motivating factor in participation in the survey. For some, this assistance was simply directed to other people or their community, “I wanted to participate in the survey to help others”. Other responses noted specifically a desire to assist those struggling with mental health conditions, “I think is important to try to help with anything that may assist others suffering suicidal thoughts and mental health issues”.

Some participants felt they could help other people due to their own experience with mental health concerns, with some aiming to use their negative experience to create a positive experience for someone else, “I don’t have a lot of time or money but would help where I can to stop others from feeling the pain I have felt”.

Other participants indicated their motivation was a desire to help but did not specify a recipient of that help. These responses were typically very broad, participants noted being “just happy to help” and wanting to “help make a difference”. One participant commented that they were motivated to participate in the survey as it was a way for them to help without negatively impacting their own wellbeing. 

Being hopeful that their contribution would be of help was also commonly reported, suggesting that while motivation was altruistic, participants were not certain that their contribution would be beneficial. Responses were similar to those mentioned above, with some participants expressing a vague hope to help “I hope I can help”, wanting to help others, “Hope it helps someone”, or specifically people struggling with poor mental health, “I hope the results of this survey go on to help improve the lives of anyone who has felt or feels suicidal”.

An external motivation source was also noted in the responses. Requests from friends influenced participation in some cases, “a friend asked me to participate so I said yes”. Similarly, a positive experience with, or the reputation of, the organisation conducting the research encouraged people to take part in the survey. 

“I participated because of my daughter’s experience and the help she received from Black Dog [Institute]. Any help my participation might provide is the least I can do”.

One participant noted that simply seeing the advert had requested participants in their area was enough to motivate them to complete the survey. 

Section summary: altruism was a considerable driving force for motivation to participate in the survey. Many people believed, or were hopeful, that their input would be an effective way improve the wellbeing of others. 

#### 3.2.2. Solve Systemic Problems

Participants identified several systemic problems they had experienced as being inherent in Australia’s mental health care system, which were not due to a specific individual or isolated factor. Responses indicated that participation was motivated by a desire to help address these problems.

Participants recognised four challenges that surround suicide: lack of public awareness about suicide, negative attitudes (stigma) shared by the community toward suicide, inadequate mental health care services, and insufficient suicide prevention programs. 

Many forms of awareness were discussed by participants. For some, they were motivated to participate to help create more awareness about avenues of support for those who may be experiencing suicidal thoughts, “I hope my contribution to this survey helps to promote the services available in my community to people struggling alone”.

Participants also discussed wanting to help raise awareness among the general population to facilitate communication so issues around mental health “can be talked about openly”, hopefully increasing people’s compassion for those struggling with suicide. 

A need to improve attitudes toward suicide was a motivator for participants; many considered stigma to be a contributor to reduced engagement with support options, particularly among men. Others considered improving attitudes and reducing stigma to be essential to improving education around mental health and suicide, result in more conversations and ultimately lead to more widespread prevention campaigns. 

“I was motivated to participate because I believe the stigma surrounding mental health needs to be eradicated. I believe if we normalise positive mental health practices and conversations like we have other health issues such as skin cancer (slip slop slap campaign), obesity and nutrition, and smoking, then the number of consumers needing to access mental health resources will be less fearful or cautious in doing so”.

Many discussed the high rates of suicide in their communities and were motivated to participate in the hopes that it would aid suicide prevention strategies. Many saw their participation as an avenue to reduce the number of deaths by suicide; many people shared the sentiment that “one person suiciding is too many”. Participants saw the impact of suicide death on the families and community and as such wanted to help reduce the number of deaths by suicide. 

“I participated due to high rate of families that seem to be dealing with this sad and devastating reality. Anything that can help understand and hopefully stop this should be supported”.

Others outlined possible prevention strategies that they hoped their participation would generate, such as more education and increased access to support services. 

Some participants reported there were simply too few services available in their communities, leaving those vulnerable to suicide without adequate support. Access to enough mental health clinicians and community support services were the most frequently discussed by participants, who saw their participation as a way to advocate for increasing the services available. 

Many participants noted that their participation was motivated by a desire to improve services in the community. Mental health services, including community-based services, access to mental health practitioners, and hospital services, were considered to be inadequate by many participants. Some believe these services required more resources and funding and hoped that by participating, these necessary resources would be allocated the areas of need. Individuals noted that poor financial situations could have detrimental impacts on ability to access services. 

“I am participating because I very strongly believe mental health in Australia needs to be far better resourced and taken far more seriously by both the government and society as a whole. I would like it if everyone had access to the same resources for more robust mental health that I was able to access. (…) To me this is completely unacceptable that anyone in our current society should be unable to seek the help required in a timely fashion due purely to their economic position”.

Many participants recognised the importance of taking part in mental health research to assist researchers to “collect a wide range of experiences” and “valuable insight”, assisting researchers to build a deeper understanding of suicide. Similarly, participants were motivated by a desire to contribute to the understanding of suicide, as they believe that the existing knowledge is insufficient, and that is contributing to high rates of suicide. In some instances, this was by sharing their personal experience of suicide:

“I want to be able to help other people going through similar things and I hope that by participating in this research I will be allowing a greater understanding of suicide to be attained”.

While others believed that their contribution to the knowledge would help the community find more effective ways of addressing suicide. 

“…my motivation for participation is that knowledge is power, and when we can get an accurate picture of what suicidality [is] in Australia (and the world) means, we can tackle it”.

Interestingly, this theme contained the only negative response toward suicide in more than 530 responses. This participant considered suicide to be unacceptable and selfish, and noted their motivation was to add this view to the current understanding of suicide.

Section summary: solving systemic problems associated with suicide was a clear motivator for participants, who saw participating in mental health research as a platform to sharing their understanding and experience and address deep-rooted, systemic problems within Australian society. 

#### 3.2.3. Lived Experience

For many participants, motivation to participate came from personal experience with mental health concerns, their own experiences with suicide, and in some cases, loss of a loved one to suicide. This differs from altruism as the motivation involves a stronger focus on making sense of their own experiences or making the lives of those struggling around them easier and safer. 

Several comments indicated that participation was due to the research topic aligning with lived experience, “I took the survey because I have experience with suicide”. Experience with mental health issues was a strong motivator for individuals. Many indicated that it was the simple alignment of the research topic with their own lives that encouraged participation, “I wanted to participate in this survey because I have a history with severe mental illness”.

Some felt they had gained useful information in overcoming their mental health challenges and wanted to share this, “I feel I have come a long way [from] where my mental health was so if I can help shed some light on my path I will”.

A similar pattern emerged for participants with a history of suicidal thoughts and attempts. Individuals expressed motivation to participate due to alignment with their experience. However, individuals with suicidal lived experience were also strongly motivated due to the depth of their understanding of a suicidal experience. Many spoke of wanting to share this experience to inform others, while others were motivated because they felt as though they had seen both sides of the experience, 

“Motivation? I used to believe that people who took their own lives were weak, and cowards. But I know(sic) understand that they see ending their lives as the ONLY answer to completely overwhelming situations where they have no control, and really believe that their family and friends are better off without them. I’ve been in this dark place many times”.

Other participants with personal experience of suicide discussed having negative emotions tied to their thoughts, and were participating as a way to deal with the thoughts in a positive way, “My main motivation for participating is the fact that I still deal with … guilt for the way I felt and the things I thought or believed”.

Participants indicated that personal experience with mental health concerns and suicidal thoughts and attempts was a motivator for participation. Participants felt they had acquired important information as a result of this experience and were eager to ensure that information was used by others.

Individuals reported that motivation came from having a loved one with mental health or suicidal struggles. Numerous individuals reported participating in the hopes that their contribution would lead to change, which would benefit their loved ones. This was particularly true for parents who suspected that their children were struggling with mental health issues and suicidal behaviours or thoughts. 

“Family members of mine are affected by mental health issues and their wellbeing is very important to me”.

“I have friends who I think are a bit depressed and I care about them and want good resources available to them”.

Knowing someone who struggles with mental health and suicidal thoughts or behaviours increased participants’ awareness of suicide and mental health. This increased awareness influenced participants to take part in the research.

“Suicide prevention has become more important to me since my brother attempted it earlier this year, which motivated me to do the survey”.

Lived experience through loved ones also leads to individuals feeling strongly about mental health and suicide, ultimately fuelling their motivation to participate in research on the topic. 

“Motivation = my husband has been suicidal this year. It’s an issue I feel strongly about”.

Participants noted that having experienced the mental health system drove them to want to see changes made, which influenced their decision to take part in the survey:

“I have had mental health issues in my family and feel the current system needs to change so felt I should participate in the study”.

Some participants reported that feelings of guilt had encouraged them to participate. Feelings of regret and remorse for not having taken action to help the individual who had suicided or where experiencing suicidal thoughts or behaviours resulted in participants engaging with the survey in an attempt to address those feelings.

“I feel a lot of guilt from my husband passing due to suicide (I could have done more?) and I want to do something positive”.

Bereavement by suicide was a motivating factor for participation in this study. Participants reported having lost family, friends, colleagues, and acquaintances, although loss of a family member was the most frequently mentioned. The majority of participants noted the loss of a loved one by suicide was the motivation for participation without further elaboration, “Motivation was the unexpected death of my brother”. Other participants said the impact of the death on themselves and their family motivated them to participate in the survey, “My main motivation was the terrible impact my son’s suicide had on our family”. 

Participants indicated that as a result of losing someone to suicide, they had obtained valuable information about how suicide affects people and saw participation in the survey as a way to share that knowledge. Others wanted to stop other people having to go through the experience of losing someone to suicide, “My nephew took his life just 2 weeks ago and I would like try and help in any way so that no other family hurts like we do at the moment!”.

Participating in memory of a loved one was also mentioned frequently, particularly in reference to family, partners, and friends. 

Working in health care or educational professions, as well as current or previous training in mental health care, and therefore having knowledge of—and exposure to—the importance of the research topic, motivated participants to be involved. 

“I worked for 42 years in the mental health field before retiring and was interested in the survey”.

“Happy to be part of any research that throws light on the issue of suicide. My profession is overrepresented in suicide statistics”.

Section summary: lived experience of suicide is a strong and unique motivator for participation in suicide prevention. Responses suggest that the emotional toll of being personally impacted by, or a loved one of someone struggling with, suicide plays a large role in the motivation to participate in this research. 

#### 3.2.4. Seeking Personal Insight

The final global theme for motivation was personal benefit, which manifested as individuals reporting that they participated as a way to process their own psychological experiences and/or understanding mental health research for their own personal growth. This was the only theme which represented solely intrinsic motivation rather than a motivation that was also influenced by external factors. 

Some participants used the survey as a tool to explore their own mental health, to see how their current state compares when exploring other possible thought patterns and behaviours. 

“… sometimes when it’s laid out like this it’s easy to see how messed up I am”.

Others reported that motivation came from a need to get their “own thoughts out”, as they were not sharing information about their current mental health state with the people in their lives. For one individual who reported wanting to share their thoughts, the participation in the survey encouraged them to speak with others about their negative thought patterns. One participant simply noted “self-improvement” was their motivation for participating. Individuals also talked of using survey participation to help them understand the mental state of loved ones who died by suicide, or to help them process their own thoughts and feelings that came about as a result of the suicide. 

“I opened this survey only because my darling grandson [died by] suicide in December and I thought there might have been something to explain it—he was beautiful”.

Participant interest in, and curiosity around, suicide research was another driver to participation, with individuals using participation in the survey to explore an area that piqued their interest. Many people reported being “mainly curious” about the process and questions involved in a suicide questionnaire. 

Interest in research and suicide also motivated individuals to complete the survey. For some, this was to explore different aspects of suicide, as they recognised that their personal experience influences their opinions on the subject.

“My own circumstances inform my thinking on suicide in general; hence my interest in completing your survey”.

Participants also reported being inquisitive about the results of the survey and took part to gain access to the findings, “I’m mostly interested in what the results show”. One participant reported choosing to complete the survey “to see how out of touch” researchers are with the experience of suicide.

Section summary: in contrast to the previous three themes, this theme showed a motivation for participation that was self-centric. This theme demonstrated the role of mental health research as a possible avenue for individuals to explore their own experiences of mental health, or to help individuals explore curiosities and areas of interest. However, this theme was smaller than the other three themes, which may indicate that self-centric motivations may not be as strong as those that serve to benefit others. 

## 4. Discussion

This study aimed to explore the motivations for participating in suicide research in a large community-based sample. Our analysis identified four key themes: altruism, desire to solve system problems, lived experience of suicide or mental illness, and seeking personal insight as key motivators for participation in this survey. This research builds markedly on previous studies, which have often only explored single aspects of the themes found here, such as lived experience [11,13,14,17] or the impact of altruism on research participation [17,31]. The thematic approach adopted for this study was able to provide a more thorough understanding of motivation to participate in a community-based sample, which also included people without any lived experience of suicide. The diversity of personal experience among those canvassed across the community who responded to a survey which did not stipulate specific experience for inclusion, likely accounted for the identification of four global themes. 

Our theme of altruism was consistent with previous research in which the desire to help others was a driving motivational factor [11,14,16,17]. People who were motivated to help due to having known someone who had died by suicide, described a variety of relationships, ranging from immediate family to personal acquaintances more broadly, yet the loss of a close loved one was the most common. This may have increased the likelihood of altruistic motivations, as one study exploring the motivations of bereaved family members found those who were immediate family members of the deceased had more altruistic motivation to participate compared to more distant relationships (e.g., extended family) [14]. Our findings demonstrate the complexity and diversity of lived experience, and importantly, the need for support for people who have been bereaved by suicide, whether an immediate family member or other connection [32]. 

Motivation also came from a hope to leverage people’s lived experience of suicide to aid suicide prevention by increasing awareness, decreasing stigma, and providing information to develop new suicide prevention resources. Some respondents felt their participation in the study would help them to process their own thoughts and feelings of suicide or of losing someone to suicide. This supports the findings of previous research in which participants found the experience of participation positive due to the ability to work through emotions relating to their suicide experience [14,33]. Mental health and suicide research may provide opportunity for participants to gain an understanding of their personal experience/s that may not be available to them otherwise. This may be through exposure to psychological language and concepts and questions that provoke insightful consideration [17].

The quality of the data captured in the optional open-text question on this survey was comprehensive, allowing us to draw meaningful conclusions about this sample’s motivations to participate in suicide research. This suggests that open-text fields in otherwise structured surveys are a feasible and potentially less resource-intensive way to capture qualitative responses, as opposed to interviews. However, we note that open-text fields are unlikely to be a suitable replacement for interviews when capturing a person’s full experience of suicide, but rather an effective option for single specific and targeted questions (such as motivation to participate, or survey feedback). Additionally, the digital delivery format of this survey may have enhanced the quality of responses due to anonymity of participants increasing their likelihood to share more openly, as one recent study found that participants were more comfortable disclosing when using digital platforms than in a face-to-face setting [34]. 

Although this study focused on the motivations of a community-based sample, our final sample had a significantly higher rate of lifetime mental health diagnosis and most of our final sample indicated they had some level of lived experience of suicide. Despite the high rate of mental health diagnosis, the experience of suicide was very diverse within our sample: personal lived experience of suicidal thoughts or behaviours, experience of watching a loved one or close friend struggle with suicidality or severe mental illness, loss of loved ones, friends, colleagues to suicide, watching family and friends struggle with the loss of a loved one, or facing the reality that their chosen profession had a high rate of suicidal behaviour. This reaffirms that suicide has wide-reaching repercussions right across communities, often impacting a large number of people [18]. This finding also highlights the importance of ensuring that future research and suicide prevention strategies continue to look beyond the usual samples. While our study sample was not representative of the general community population (i.e., underrepresentation of men) [7], the recruitment of this sample was not specifically directed at individuals with lived experience, as typically seen in studies exploring motivation to participate in suicide research [11,13,14,17], and the novelty of our findings suggest there may be more to learn from more diverse samples.

We found that many were motivated to participate due to an exposure to suicide that included a range of connections to the suicidal person, including intimate partners, extended family, and close and distant acquaintances. While this exposure to suicide may be increasing an individual’s interest in the study topic, and therefore may increase the likelihood that such an individual self-selects into the study [35,36], lived experience of many of our participants was still more diverse (i.e., close and distant acquaintances) than the lived experience typically examined in suicide prevention research, for instance suicidal individuals or bereaved family or carers [37]. As such, providing opportunities to those across the community to engage in suicide research has resulted in a sample showing the extent of awareness and impact from the phenomena of suicide and suggests a need for broad suicide prevention strategies that involve people not typically identified as “at risk groups”.

There was a small cohort of participants who reported experiencing suicide and mental health issues but did not want to disclose this to others; as such, there is potential for community-based samples to be a useful first step in tapping into this population who may be at the greatest risk of not accessing help. Online surveys may provide the anonymity that the participant desires when disclosing suicide or mental health issues [38], however, this puts researchers in a difficult ethical situation. To mediate this effect, phone numbers for a variety of mental health and suicide helplines were provided at the beginning and end of the survey, however, use of these helplines is unknown. Future researchers may need to consider such disclosure when exploring suicide in community-based populations. Understanding their motivations to participate may help identify and develop strategies to engage individuals in support programs and services to address their suicidality and/or mental health issues. 

Despite previous studies finding that participating in suicide research does not increase people’s risk [39,40], this remains a significant concern of research ethics review boards due to risks of psychological harm. In the present study, wanting to help others, both at an individual and structural level, emerged as a strong theme in participants’ motivations. Previous research has suggested that engaging in altruistic behaviour makes participation in suicide research a generally positive experience [15,33,41]. This should be considered by ethics committees when weighing the risks and benefits to participation in suicide research. Similarly, we found evidence that participating in this study provided an opportunity for some individuals to start processing negative emotions relating to the experience of suicidal thoughts or of caring for someone with suicidal thoughts. These findings, combined with the benefits of altruism, may help to support decision making processes in ethics committees about the appropriateness and potential benefits of involvement in suicide prevention research.

This study had some limitations. The question regarding motivations was optional, and therefore self-selection bias may have influenced who responded. Further, we only asked about motivations using a single question, so were not able to obtain more details about participants’ responses. This may have been useful in providing a richer understanding of the depth and variability of people’s motivations, given the large sample size. Nevertheless, the richness of analysis enabled through use of TNA identified several different types of motivations, many of which are consistent with existing research which utilised more detailed and resource-intensive qualitative interviewing methodology [13,16]. Additionally, recruitment for this study was through social media, and, as such, might limit the accessibility of the survey to many in the community and may be impacted further using only a subset of social media platforms (Facebook and Instagram) [5]. As such, caution is advised when making any generalisations about applicability to the general population. Lee and Torok et al. [7] discuss potential strategies to improve representativeness of samples recruited through social media. Similarly, the advertisements used for recruitment focused on a call to action which encouraged individuals to help others through participation. This may have resulted in more people who are altruistically motivated participating in this study. It should also be noted that the sites selected for the LifeSpan trial were selected due to the high proportion of suicides in those areas; this may contribute to the high level of lived experience reported by participants. 

Further research should also explore the experiences of suicide across the general population, as this sample has identified unique experiences that highlight the important role of generalized samples of the broader community in suicide research to uncover novel or more representative experiences of suicide. Additionally, more exploration of participation motivations would be beneficial among hard-to-reach sub-groups such as male, older, and culturally diverse populations. We also find, as previously discussed by Hjelmeland and Knizek [42], that more qualitative research is necessary to broaden our understanding of suicide, and novel and unique approaches to qualitive research should be utilised in order to ensure greater breadth to our understanding. 

## 5. Conclusions

Researchers need to continue to find novel approaches to improve and strengthen the understanding of suicide, and therefore optimise suicide prevention initiatives. This study reveals the benefit of short-answer responses within structured surveys and the importance of engaging new populations, such as the general community, in suicide research. We found that the impact of suicide extends well beyond those immediately connected to a suicidal individual, as well as supporting existing research showing that altruism is a considerable motivation for participation in suicide research. By giving a voice to a general community sample, we have shown the importance of continuing to include broader samples in future suicide research to ensure that all avenues for potential suicide prevention are exposed. 

## Figures and Tables

**Figure 1 ijerph-18-04705-f001:**
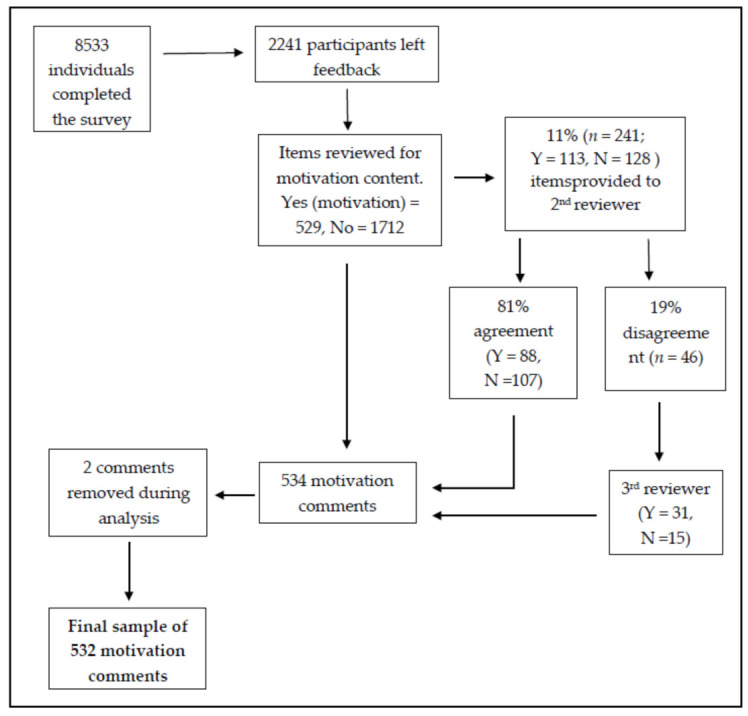
Study participant flow diagram.

**Figure 2 ijerph-18-04705-f002:**
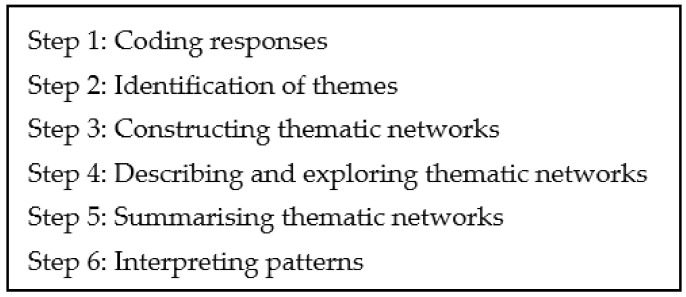
Thematic Network Analysis steps (Attride-Stirling, 2001).

**Table 1 ijerph-18-04705-t001:** Demographic and mental health characteristics of respondents who left a motivation comment compared with all respondents.

		Motivation Comment	All Participants	Motivation Comments Versus All Participants	
		*n* (%)	*n* (%)	Chi-Square (df, *n*)	*p* Value
**Gender**					
	Female	344 (65.0)	5675 (67.2)	1.08 (df = 1, *n* = 8972)	0.299
	Male	185 (35.0)	2768 (32.8)		
**Age**					
	18–34 years	140 (26.5)	2413 (28.6)	4.51 (df = 2, *n* = 8977)	0.105
	35–59 years	299 (56.6)	4882 (57.8)		
	60+ years	89 (16.9)	1154 (13.7)		
**Relationship status**					
	Never coupled	133 (21.2)	2012 (23.8)	2.05 (df = 2, *n* = 8996)	0.359
	Coupled	321 (60.3)	4871 (57.6)		
	No longer coupled	98 (18.4)	1581 (18.7)		
**Employment status**					
	Full-time	220 (41.4)	3836 (45.3)	3.53 (df = 2, *n* = 8991)	0.172
	Part-time	144 (27.1)	2228 (26.3)		
	Unemployed	167 (31.5)	2396 (28.3)		
**Education**					
	School-based	136 (25.7)	1860 (21.9)	5.48 (df = 2, *n* = 9024)	0.065
	Undergraduate	327 (61.7)	5343 (62.9)		
	Postgraduate	67 (12.6)	1291 (15.2)		
**Previous mental health diagnosis**					
	No	148 (27.8)	2819 (33.2)	6.55 (df = 1, *n* = 9025)	0.010 *
	Yes	384 (72.2)	5674 (66.8)		

Note: * indicate values <0.05. df = degrees of freedom.

**Table 2 ijerph-18-04705-t002:** Thematic Network Analysis and resultant thematic networks.

Network	Global Themes	Organising Themes	
1	Altruism (*n* = 187)	Help Other People	Participated due to wanting to help other people or the community, or specifically people with mental health concerns. Some wanted to use their personal experience to help others.
		Desire to Help	Broad desire to help with no specific target and considered the survey an easy way to provide help.
		Hopeful to Provide Help	Hopeful contribution will be of help, to other people, or to no specific target.
		External Motivation Source	Participated due to a friend’s request or because of a positive association with the institute conducting the research.
2	Solve Systemic Problems (*n* = 195)	Want to Solve Identified Problem	Participants saw partaking in the survey as a method to help address problems such as the need to raise awareness, to change attitude and stigma around suicide, assist suicide prevention, and increase and/or improve services.
		Add to Suicide Knowledge	Adding to the knowledge and understanding of suicide, and the role of assisting researchers in that, were motivators. One participant was motivated by a desire to share their negative view of suicide.
3	Lived Experience (*n* = 229)	Personal Experience	Personal experience with suicide attempts, suicidal thoughts, and mental health motivated participants through connection to the survey content.
		Loved One with Personal Experience	Experience of a loved one with suicidal thoughts and actions motivated participants through a desire to ensure resources were available to help their loved ones. Having a loved one with mental health issues, as well as suicidal thoughts or behaviours, created a stronger awareness in participants, encouraging their participation.
		Bereaved by Suicide	Knowing someone who has died by suicide drove motivation to participate through participants increased knowledge and strong emotional reactions to the loss.
		Professional Experience	Working in a mental health profession or support role, as well as being exposed to training or education in suicide, increased participant awareness of suicide and motivated participation. Employment in a profession at high risk of suicide also acted as motivation due to the increased awareness of the impacts of suicide.
4	Seeking Personal Insight (*n* = 23)	Intellectual Inquisitiveness	Participant interest in and curiosity about mental health research, the process involved, and the nature of the questions asked.
		Clarity of Oneself	Using the participation in the survey as a way of making sense of individual’s own thoughts or experiences with suicide. Participation was also seen as an avenue to self-improvement.

Note: *n* refers to the number of responses; this exceeds the number of participants as some participants discussed multiple motivations for participation.

## Data Availability

Data available upon request from the corresponding author. The data are not publicly available due to privacy restrictions.

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
