# Peer review of "A Qualitative Analysis of Motivators to Participation in Suicide-Focused Research from a Community-Based Australian Sample"

_ijerph, 2021, doi:10.3390/ijerph18094705_

Round 1

Reviewer 1 Report

The topic of the paper is applicable across a range of types of suicide research beyond participation in survey methods and is thus of high relevance in general. The qualitative methods were conducted and explained well. In general, the writing was also of high quality, though there are minor errors that would benefit from copy-editing. The point that is made near the end of the introduction about increasing the generalizability of research samples is astute and could be given more emphasis. Questions related to external validity may be key to improving the eventual prevention or intervention outcomes that stem from research. There are multiple instances where the argument is made that increasing participation rates is “essential” for improving suicide prevention, but few that answer why that is essential.

The note that is included in the discussion about the potential benefits for participation (such as altruism) could be more strongly highlighted as a note to review boards that traditionally focus exclusively on risks of suicide research.

There are at least two aspects of the study procedure that would provide important context for the results, but are not accorded much attention. All motivations for participating in the study were in response to the study advertisement, but there is no information provided about what text, images, or other media were included in the ads. No information is provided about parameters used for audience selection for the ads. No information is provided about the conversion rate for the ad (i.e., what proportion of individuals who had sufficient motivation to reach the survey landing page proceeded to the next step of taking the survey, and what proportion of individuals who started the survey completed it). There is a brief note about limitations associated with recruitment through social media (lines 550-552). However, the advertisement was used in a subset of social media (Facebook and Instagram), which presents an additional limitation to generalizability.

While the paper indicates that online surveys have “been shown to have similar representativeness to traditional methods” and may be “effective for reaching hard to reach populations” (lines 43-45) it cites only one study from 2014. There are others who state that online opportunities tend to exclude individuals from under-privileged areas and thus create a socioeconomic bias.

The introduction notes that “certain populations, such as men, older adults, and culturally and linguistically diverse minority groups are less likely to participate in research, but have a higher risk of suicide” (lines 72-76). The study does not report on racial, ethnic, cultural, or linguistic diversity of the sample. Consistent with other research, in this study men are under-represented. Older adults were slightly less likely to respond to the open-ended question. It was useful to see that the paper addressed questions about whether or not the sub-sample was representative of the general sample, but it does not address the issues with generalizability that are mentioned at the outset. No sub-group analysis is presented that might examine the participation motivations of the sub-groups that are noted as important.

Author Response

Response to Reviewer 1  Comments

Thank you for taking the time to review this manuscript and for the detailed and considerate comments and recommendations you provided. we have made changes as suggested, as detailed below. You will find the reviewer comments in bold and italicised. Our responses are in red text.

Point 1: The topic of the paper is applicable across a range of types of suicide research beyond participation in survey methods and is thus of high relevance in general. The qualitative methods were conducted and explained well. In general, the writing was also of high quality, though there are minor errors that would benefit from copy-editing.

 Thank you for this feedback, we have reviewed the manuscript and made a number of typographical and grammatical edits.

Point 2: The point that is made near the end of the introduction about increasing the generalizability of research samples is astute and could be given more emphasis. Questions related to external validity may be key to improving the eventual prevention or intervention outcomes that stem from research. There are multiple instances where the argument is made that increasing participation rates is “essential” for improving suicide prevention, but few that answer why that is essential.

Thank you for this suggestion. We have added an additional sentence to the introduction to further indicate why increasing participation is important to improving suicide prevention initiatives; “By increasing community participation in suicide prevention research, study findings will have greater external validity which may enable improvements in, or refinement of, suicide prevention efforts.” [Lines 44-46]

Additionally, we have amended the abstract to make the ‘why’ clearer – “Increasing the rates of general community participation in suicide research may improve knowledge generalisability as it relates to suicidal behaviour and lead to new suicide prevention approaches.” [Lines 16-19]

We have also given more weight to this point by amending the following sentence in this discussion; “Further research should also explore the experiences of suicide across the general population, as this sample has identified unique experiences that highlight the important role of generalised samples of the broader community in suicide research to uncover novel or more representative experiences of suicide.” [Lines 595-600]

Point 3: The note that is included in the discussion about the potential benefits for participation (such as altruism) could be more strongly highlighted as a note to review boards that traditionally focus exclusively on risks of suicide research.

This section has been amended with the addition of the following sentence: “This [altruism] should be considered by ethics committees when weighing the risks and benefits to participation in suicide research.”. [Lines 568-569]

The final sentence in that section has also been amended to further highlight the potential benefits to participating in suicide research; “These findings, combined with the benefits of altruism, may help to support decision making processes in ethics committees about the appropriateness and potential benefits of involvement in suicide prevention research.” [Lines: 572-574]

Point 4: There are at least two aspects of the study procedure that would provide important context for the results, but are not accorded much attention. All motivations for participating in the study were in response to the study advertisement, but there is no information provided about what text, images, or other media were included in the ads. No information is provided about parameters used for audience selection for the ads. No information is provided about the conversion rate for the ad (i.e., what proportion of individuals who had sufficient motivation to reach the survey landing page proceeded to the next step of taking the survey, and what proportion of individuals who started the survey completed it).

Additions have been made to the methods section to provide more clarity around the content of the advertisements; “Advertisements were both gender neutral and male targeted. The gender-neutral adverts contained phrases such as “Lend a Voice to Suicide Prevention” and gender-neutral imagery. Male targeted adverts displayed male focused imagery, used male pronouns, and contained phases such as “voice of local blokes to help make it better”. Ads were targeted at individuals who met the inclusion criteria; Individuals accessing these social media platforms were eligible for inclusion if aged 18 years or over and living in a trial or control region of the NSW LifeSpan suicide prevention trial.” [Lines 114-120]

A sentence has been added indicating that detailed information about the ads and the conversion rates for the ads can be found in the study conducted by Lee and Torok et al., 2020; “More information about the advertisements and the conversion rates of the advertisements can be found in Lee & Torok [7].” [Lines 121-122]

A note was also added indicating that 77.8% of all survey attempters completed the survey. [Line 142]

Implications of the content of the advertisements have been added to the discussion; “Similarly, the advertisements used for recruitment focused on a call to action which encouraged individuals to help others through participation, this may have resulted in more people who are altruistically motivated participating in this study.” [Lines 588-591]

Point 5: There is a brief note about limitations associated with recruitment through social media (lines 550-552). However, the advertisement was used in a subset of social media (Facebook and Instagram), which presents an additional limitation to generalizability.

We have noted the limitation of using only a subset of social media which may impact generalizability of the population recruited as recommended; “Additionally, recruitment for this study was through social media, and as such might limit the accessibility of the survey to many in the community and may be impacted further using only a subset of social media platforms (Facebook and Instagram) [5].” [Lines 583-586]

Point 6: While the paper indicates that online surveys have “been shown to have similar representativeness to traditional methods” and may be “effective for reaching hard to reach populations” (lines 43-45) it cites only one study from 2014. There are others who state that online opportunities tend to exclude individuals from under-privileged areas and thus create a socioeconomic bias.

Thank you for bringing this issue to our attention, we have added this information accordingly to the paragraph; “While online surveys have been shown to fail to engage subsamples of the general population, such as  participants with lower socioeconomic status due, in part, to computer illiteracy or poor access to technology [4, 5], online surveys have been shown to have similar representativeness to traditional research methods in respect to age, gender, education, and cultural diversity… ” [Lines 50-54]

Point 7: The introduction notes that “certain populations, such as men, older adults, and culturally and linguistically diverse minority groups are less likely to participate in research, but have a higher risk of suicide” (lines 72-76). The study does not report on racial, ethnic, cultural, or linguistic diversity of the sample. Consistent with other research, in this study men are under-represented. Older adults were slightly less likely to respond to the open-ended question. It was useful to see that the paper addressed questions about whether or not the sub-sample was representative of the general sample, but it does not address the issues with generalizability that are mentioned at the outset. No sub-group analysis is presented that might examine the participation motivations of the sub-groups that are noted as important.

We agree that further analysis of the motivations to participate of the noted sub-groups would be beneficial, unfortunately, ethnic, cultural and linguistic diversity was not assessed in the survey. Additionally, regrettably the process of the qualitative analysis did not allow us to link the thematically coded responses back to individual participants, therefore did not allow us to do any further analysis of racial, age, or gender specific motivations.

We have added a sentence to the discussion highlighting that the sample is not representative of the general population, but noting that it is still markedly more diverse than much of the suicide research conducted to date: “While our study sample was not representative of the general community population (i.e. underrepresentation of men) [7], the recruitment of this sample was not specifically directed at individuals with lived experience, as typically seen in studies exploring motivation to participate in suicide research [11, 13, 14, 17], and the novelty of our findings suggest there may be more to learn from more diverse samples.” [Lines 529-534]

We have also noted in the discussion, that further analysis of the motivations of these specific subgroups would be beneficial going forward; “Additionally, more exploration of participation motivations would be beneficial among hard-to-reach sub-groups such as men, elderly, and culturally diverse populations.” [Lines 598-600]

Reviewer 2 Report

Thank you for the opportunity to review this manuscript. I found it well-written and appreciated the thoroughness and clarity of the authors’ presentation. The topic of increasing suicide research participation to better inform our understanding of the phenomenon is crucially important and very timely. While my overall impression of the manuscript is very positive, there are some suggestions I have outlined that I believe would improve the paper and I would encourage the authors to consider these in their revision.

Title:

  • I might suggest a slight change from “suicide prevention research” to “suicide-focused research” since it is not clear from how the survey is described that the intention was to prevent suicide or develop prevention programs. However, perhaps this was a larger goal of the parent study and could be elaborated upon further in the manuscript to clarify.

Introduction:

  • Page 1: “There has not been a significant shift in the number of people dying by suicide…” Given suicide rates have increased in some places/countries/groups and decreased in others, it may be clearer to say that “there has not been a global reduction in suicide deaths” or something similar.

Methods:

  • What was the focus of the recruitment ads? That is, since the rest of the context suggests the study was advertised as being suicide-related, how might this have affected who participated and who did not?
  • Since the authors tout this as a community sample, it would be of interest to see how the sample compares to the population from which participants were recruited. That might further speak to whether the participants represent the community, above and beyond what effect specific recruitment language/framing might have had.

Results:

  • It would be beneficial to compare participants who left a comment to those who did not – rather than those who did compared to the full sample (of which they are also a part)?
  • On page 11: “This [seeking personal insight] was the only theme which represented intrinsic motivation rather than an influence of external factors.” I would argue that wanting to help others by sharing information and perspective is also somewhat intrinsically motivated… similarly for those who had lost someone or wanted to share their lived experience, might that not also be “intrinsic” motivation? What is the rationale for how the authors have chosen to designate “external” vs. “intrinsic” motivation?

Discussion:

  • On page 13: “Although it would be expected that people who self-select to participate in a study on a particular topic would have greater interest in that topic, our participants were predominantly driven by a breadth of lived experience, rather than a general personal or professional interest in suicide prevention.” I don’t think the authors have enough information to make this claim. First, only a subset of participants provided explanation for their reason for participating. Second, only one participant expressed negative attitudes toward suicide, suggesting there is a bias in the sample toward more compassion/understanding/curiosity toward the topic. And third, a breadth of lived experience still suggests a greater interest than might be expected for people without such lived experience.
  • On page 14: “We found evidence that participating in this study helped individuals to process negative emotions relating to the experience of suicidal thoughts…” This is not how the results were presented, so I think clarity either in the results or in this discussion is needed. From the results, it seems that participants chose to participate in order to seek these benefits, but they did not necessarily report that they OBTAINED those benefits. And since there was no pre-post assessment of negative emotions (for example) as a result of study participation, this is overstating the findings.

Minor comments:

  • N and % of total sample with a lifetime mental health condition diagnosis is missing from in-text summary on page 5
  • Table 1 formatting is inconsistent

Author Response

Response to Reviewer 2 Comments

Thank you for taking the time to review this manuscript and for the detailed and considerate comments and recommendations you provided. we have made changes as suggested, as detailed below. You will find the reviewer comments in bold and italicised. Our responses are in red text.

Point 1: I might suggest a slight change from “suicide prevention research” to “suicide-focused research” since it is not clear from how the survey is described that the intention was to prevent suicide or develop prevention programs. However, perhaps this was a larger goal of the parent study and could be elaborated upon further in the manuscript to clarify.

We agree that “suicide-focused” is a better descriptor and have modified the title accordingly.

Introduction:

Point 2: Page 1: “There has not been a significant shift in the number of people dying by suicide…” Given suicide rates have increased in some places/countries/groups and decreased in others, it may be clearer to say that “there has not been a global reduction in suicide deaths” or something similar.

We agree that our original wording was unclear, we have amended the sentence as suggested; “Despite some countries making progress toward suicide prevention, globally there has been no significant reduction in the number of suicide deaths over the last few decades” [Lines 36-38]

Methods:

Point 3: What was the focus of the recruitment ads? That is, since the rest of the context suggests the study was advertised as being suicide-related, how might this have affected who participated and who did not? Since the authors tout this as a community sample, it would be of interest to see how the sample compares to the population from which participants were recruited. That might further speak to whether the participants represent the community, above and beyond what effect specific recruitment language/framing might have had.

We have updated the methods section to provide more context around the nature of the advertisements.

“Advertisements were both gender neutral and male targeted. The gender-neutral adverts contained phrases such as “Lend a Voice to Suicide Prevention” and gender-neutral imagery. Male targeted adverts displayed male focused imagery, used male pronouns, and contained phases such as “voice of local blokes to help make it better”. Ads were targeted at individuals who met the inclusion criteria; Individuals accessing these social media platforms were eligible for inclusion if aged 18 years or over and living in a trial or control region of the NSW LifeSpan suicide prevention trial. More information about the advertisements and the conversion rates of the advertisements can be found in Lee & Torok [7].” [Lines 114-122]

Analysis reported by Lee and Torok et al., indicated that this sample was unfortunately not representative of the wider community population. A sentence has been added to the methods to indicate this, as this was previously missing from the manuscript; “Analysis conducted be Lee & Torok [7] found that the sample was not representative of the wider community.” [Lines 125-126].

Additionally, we also discuss the implications of the messaging and nature of the ads in regard to any potential influence on motivations to participate in the discussion; “Similarly, the advertisements used for recruitment focused on a call to action which encouraged individuals to help others through participation, this may have resulted in more people who are altruistically motivated participating in this study.” [Lines 588-591]

Results:

Point 4: It would be beneficial to compare participants who left a comment to those who did not – rather than those who did compared to the full sample (of which they are also a part)?

We considered running this suggested analysis however we feel it is not in alignment with the purpose of this paper. The purpose of this paper is to understand what motivates people to engage in research with the intention that other researchers take that information to develop engagement strategies to hopefully encourage others to participate and increase external validity of suicide research findings. Providing the demographic/clinical profile of those who didn’t leave a comment doesn’t provide much valuable information as these individuals still participated in, and completed, the survey however just didn’t provide information about their motivations to participate.

Point 5: On page 11: “This [seeking personal insight] was the only theme which represented intrinsic motivation rather than an influence of external factors.” I would argue that wanting to help others by sharing information and perspective is also somewhat intrinsically motivated… similarly for those who had lost someone or wanted to share their lived experience, might that not also be “intrinsic” motivation? What is the rationale for how the authors have chosen to designate “external” vs. “intrinsic” motivation?

Thank you for this feedback. Our rational for how we chose to designate “external” and “intrinsic” was based on our interpretation that ideas shared in the ‘seeking personal insight’ theme did not contain any components of motivation which could be considered motivation to help anyone but them participant themselves. We agree that there is some degree of intrinsic motivation in sharing one’s own lived experience or desire to help others, however we considered the “seeking personal insight” theme to be unique in that it did not have this combination of both external and intrinsic motivations.

To provide more clarity around this in the manuscript, we have made slight amendments to the sentence to clarify that the “seeking personal insight” theme is absent of any external motivations; “This was the only theme which represented solely intrinsic motivation rather than a motivation that was also influenced by external factors.” [Lines 436-437]

Discussion:

Point 6: On page 13: “Although it would be expected that people who self-select to participate in a study on a particular topic would have greater interest in that topic, our participants were predominantly driven by a breadth of lived experience, rather than a general personal or professional interest in suicide prevention.” I don’t think the authors have enough information to make this claim. First, only a subset of participants provided explanation for their reason for participating. Second, only one participant expressed negative attitudes toward suicide, suggesting there is a bias in the sample toward more compassion/understanding/curiosity toward the topic. And third, a breadth of lived experience still suggests a greater interest than might be expected for people without such lived experience.

Thank you for bringing this to our attention, it was a notable oversight. We have amended that section of the discussion accordingly.

“While this exposure to suicide may be increasing an individual’s interest in the study topic, and therefore may increase the likelihood that such an individual self-selects into the study [35, 36], lived experience of many of our participants was still more diverse (i.e. close and distant acquaintances) than the lived experience typically examined in suicide prevention research, for instance suicidal individuals or bereaved family or carers [37]. As such, providing opportunities to those across the community to engage in suicide research …” [Lines 537-546]

Point 7: On page 14: “We found evidence that participating in this study helped individuals to process negative emotions relating to the experience of suicidal thoughts…” This is not how the results were presented, so I think clarity either in the results or in this discussion is needed. From the results, it seems that participants chose to participate in order to seek these benefits, but they did not necessarily report that they OBTAINED those benefits. And since there was no pre-post assessment of negative emotions (for example) as a result of study participation, this is overstating the findings.

Thank you for this comment. We feel as though the reporting of the results did adequately show that some participants were benefited by participating in the study as it provided an avenue to process their negative emotions as can be seen here “Others reported that motivation came from a need to get their “own thoughts out” as they were not sharing information about their current mental health state with the people in their lives. For one individual who reported wanting to share their thoughts the participation in the survey encouraged them to speak with others about their negative thought patterns.” [page 12, Lines 442-446], and “Individuals also talked of using survey participation to help them understand the mental state of loved ones who died by suicide, or to help them process their own thoughts and feelings that came about as a result of the suicide.” [Lines 447-449]

However, we note that the wording of the original statement in the discussion may have been too definitive, and we have amended the wording slightly to account for not being able to confirm if participants were actually processing their negative emotions as a result of participation;  “Similarly, we found evidence that participating in this study provided an opportunity for some individuals to start processing negative emotions relating to the experience of suicidal thoughts or of caring for someone with suicidal thoughts.” [Lines 569-572]

Minor comments:

Point 8: N and % of total sample with a lifetime mental health condition diagnosis is missing from in-text summary on page 5 – we are unsure what section you are referring to here, as the total sample N and % were reported initially in this section, however we note that the N were missing from the reporting of the chi-square analysis results and have amended accordingly. [Line 206]

Point 9: Table 1 formatting is inconsistent – thank you for bringing this to our attention, formatting has been fixed.